Comparison of blood viscosity models in different degrees of carotid artery stenosis

Liu Siyu 1
Wang Sai 2
Tian Hongan 2
Xue Junzhen 3
Guo Yuxin 1
Yang Jingxi 1
Jiang Haobin 1
Yang Jian bao yangjianbao@hbust.edu.cn 1
Zhang Yang zhangyang619@hbust.edu.cn 4
1 School of Public Health and Nursing, Xianning Medical College, Hubei University of Science and Technology , Xianning , China
2 The First Affiliated Hospital of Hubei University of Science and Technology , Xianning , China
3 Health Management Faculty, Xianning Vocational and Technical College , Xianning , China
4 Institute of Engineering and Technology, Hubei University of Science and Technology , Xianning , China
Uversky Vladimir
Electronic publication date: 2025 Apr 28
Publication date: 2025
Volume: 13
Electronic Location ID: e19336
Received 2024 Dec 2; Accepted 2025 Mar 26
Copyright: ©2025 Liu et al.
Copyright year: 2025
Copyright holder: Liu et al.
License: This is an open access article distributed under the terms of the Creative Commons Attribution License, which permits unrestricted use, distribution, reproduction and adaptation in any medium and for any purpose provided that it is properly attributed. For attribution, the original author(s), title, publication source (PeerJ) and either DOI or URL of the article must be cited.
License URL: https://creativecommons.org/licenses/by/4.0/

Keywords: Hemodynamics, Carotid artery stenosis, Carreau model, Newtonian fluid, Computational fluid dynamics

Funding: Hubei Natural Science Foundation Program 2023AFC048 Hubei University of Science and Technology PhD Start-up Fund Project BK202112 This work was supported by the Hubei Natural Science Foundation Program (No. 2023AFC048) and Hubei University of Science and Technology PhD Start-up Fund Project (No. BK202112). The funders had no role in study design, data collection and analysis, decision to publish, or preparation of the manuscript.

==============================
Background

Carotid atherosclerotic vascular disease significantly contributes to strokes, presenting a heightened risk of early recurrent ischemia. Computational fluid dynamics (CFD) has shown potential in predicting subsequent stroke recurrence in patients with carotid stenosis.

Objective

This study aims to investigate the differences in computational time and accuracy of four key hemodynamic indices—wall shear stress (WSS), time-averaged wall shear stress (TAWSS), Oscillatory Shear Index (OSI), and relative residence time (RRT)—across different viscosity models, thereby providing optimal model selection for clinical cases and offering guidance for clinicians’ decision-making.

Methods

A three-dimensional vessel model was established using computed tomography angiography (CTA), followed by CFD simulations to calculate WSS, TAWSS, OSI, and RRT. The accuracy of the simulations was validated by comparing the results with those from Razavi et al. (10.1016/j.jbiomech.2011.04.023). Numerical errors in different parameters under varying stenosis levels and viscosity models were analyzed.

Results

In the transient state, when degree of stenosis is 38%, 72%–84%, the performance difference between the two is less than 6%. For TAWSS, the difference is 0% when degree of stenosis is 12%, 18%, 26%, 54%, and 76%. For OSI, the difference is 0% when stenosis is 16%, 18%, 26%. For RRT, the difference between the two is 0% when degree of stenosis is 18% and 84%. WSS exhibited an increasing trend with higher degrees of stenosis. TAWSS demonstrated significant variation in moderate to severe stenosis, while OSI increased markedly in cases of moderate to severe stenosis. High RRT values in severely stenotic regions indicated a propensity for atherosclerotic lesion development.

Conclusion

This study systematically quantified the discrepancies between Newtonian and non-Newtonian blood viscosity models in predicting hemodynamic parameters across different degrees of carotid artery stenosis. Statistical analyses revealed significant differences between the two models in WSS, TAWSS, OSI, and RRT (p < 0.001 for all parameters). Newtonian models, while computationally efficient, overestimated TAWSS, OSI, and RRT while underestimating WSS, particularly in moderate to severe stenosis. In contrast, non-Newtonian models provided more physiologically accurate predictions, especially in regions with high shear stress variations. The results emphasize the importance of selecting appropriate viscosity models for CFD-based patient-specific risk assessment, particularly in stroke prediction, plaque evaluation, and surgical planning. Non-Newtonian models should be prioritized in high-risk patients where flow disturbances are more pronounced, whereas Newtonian models remain suitable for early screening and rapid assessments.

Introduction

Carotid atherosclerotic vascular disease is a major cause of stroke, posing a higher risk of early recurrent ischemia compared to other stroke types (Lovett, Coull & Rothwell, 2004; Tu et al., 2023). Hemodynamic analysis using computational fluid dynamics (CFD) has proven effective in predicting stroke recurrence in patients with carotid stenosis, offering insight into the relationship between hemodynamic parameters and disease progression (Politis et al., 2008). CFD simulations have been widely employed to assess variations in hemodynamic parameters under different viscosity models (Nader et al., 2019; Wang et al., 2023b). Key hemodynamic indices, such as wall shear stress (WSS), time-averaged wall shear stress (TAWSS), oscillatory shear index (OSI), and relative residence time (RRT), are critical in understanding the effects of carotid artery stenosis on blood flow (Rahman et al., 2018; Wong et al., 2020; Liu et al., 2021). WSS refers to the frictional force exerted by blood flow on the vascular endothelium per unit area of the vessel wall. TAWSS represents the average WSS over a full cardiac cycle. OSI is a dimensionless parameter that quantifies changes in the direction of WSS over time, reflecting the degree of oscillation in flow patterns. RRT reflects the duration blood remains in contact with or near the endothelium.

While Newtonian models, which treat blood as a simple fluid with constant viscosity, are commonly used for their computational efficiency (Gasecki et al., 1995; Al-Azawy, Kadhim & Hameed, 2020). However, blood is inherently a non-Newtonian. The non-Newtonian behavior of blood, characterized by shear-thinning viscosity, particularly in regions with low shear rates, is better represented by models that account for this complex viscosity effect (Bernabeu et al., 2013). As shown in Table 1, multiple studies have compared the two models in different vascular conditions, but there is still a need for a direct comparison of their performance in predicting key hemodynamic parameters in carotid stenosis. There is a significant lack of research comparing variations in WSS, TAWSS, OSI, and RRT under different viscosity models at varying degrees of carotid artery stenosis. Such comparisons analyses are crucial for evaluating internal carotid artery stenosis (ICAS) progression and formulating effective rehabilitation strategies.

Table 1 CFD study in blood vessel.

Authors	Subject	Year	Viscosity model	Parameters	Flow type	Vascula wall	
Liu et al. (2021)	ICA	2021	Newtonian non-Newtonian	WSS, PR	Laminar	Rigidity	
Lopes et al. (2020)	Blood Vessel	2020	Newtonian non-Newtonian	WSS, V	Laminar	Rigidity	
Rayz & Cohen-Gadol (2020)	Cerebral Aneurysms	2020	non-Newtonian	WSS, TAWSS, V, VO, WSSSG	Laminar	Rigidity	
Azar et al. (2019)	ICA	2019	non-Newtonian	WSS, TAWSS, OSI, RRT	Laminar	Rigidity	
Buradi & Mahalingam (2020)	CA	2020	Newtonian	WSS, TAWSS, OSI, RRT	Laminar	Rigidity	
Yao et al. (2019)	ICA	2019	Newtonian	WSS, TAWSS, OSI, WSSSG	Laminar	Rigidity	
Colombo et al. (2022)	SFA	2022	Newtonian	WSS, TAWSS, OSI, WSSR	Laminar	Rigidity	
Eslami et al. (2020)	CA	2020	Newtonian	WSS, TAWSS	Laminar	Elasticity	
Li et al. (2019)	ICA	2019	Newtonian	WSS, TAWSS, P V, VS, VV, VO	Laminar	Rigidity	
Sia et al. (2019)	ICA	2019	Newtonian	WSS, OSI, P V, VS,	Laminar	Rigidity	
Veeturi et al. (2021)	MCA	2021	Newtonian	WSS, OSI,	Laminar	Rigidity	
Nagargoje & Gupta (2020)	ICA	2020	Newtonian	WSS, OSI	Laminar	Rigidity	
Khan et al. (2020)	Aneurysms	2020	Newtonian	WSS, OSI, SPI	Laminar	Rigidity	
Wong et al. (2020)	RCA	2020	Newtonian	WSS, WPG	Laminar	Rigidity	
Rezaeitaleshmahalleh et al. (2023)	AAA	2023	Newtonian	WSS, V	Laminar	Rigidity	
Khodaei et al. (2023)	Cardiac Value	2023	Newtonian	TAWSS, V	Laminar	Rigidity	
Wang, et al. (2023a)	OA	2023	Newtonian	V, MFR	Laminar	Rigidity	
Vardhan et al. (2019)	CA	2019	Newtonian	ESS	Laminar	Rigidity	
Leng et al. (2019)	ICA	2019	Newtonian	WSSR, PR	Laminar	Rigidity	
Zhang et al. (2021)	MCA	2021	Newtonian	WSSR, WSS	Laminar	Rigidity	

In this study, the three-dimensional vessel models were constructed using CTA, followed by CFD-based hemodynamic simulations to calculate WSS, TAWSS, OSI, and RRT (Fig. 1). Then, the accuracy of simulation is validated by comparing the results with those reported by Razavi, Shirani & Sadeghi (2011). Finally, the calculation errors of various hemodynamic parameters in different models under distinct viscosity assumptions are compared and analyzed. This study discusses how the differences in blood viscosity models (Newtonian vs. non-Newtonian) affect the hemodynamic parameters in varying degrees of carotid artery stenosis, and the impact these differences have on clinical decision-making. It aims to evaluate the computational time and accuracy of Newtonian and non-Newtonian models in simulating WSS, TAWSS, OSI, and RRT across varying degrees of carotid artery stenosis (S). By comparing these models, we aim to identify the most suitable viscosity model for clinical use, balancing accuracy and computational efficiency. Our findings will provide guidance for clinicians in selecting the optimal model for assessing hemodynamic characteristics in patients with carotid atherosclerosis.

Figure 1 Workflow diagram.

(A) CTA image acquisition. (B) Using Geomagic to generate geometric model. (C) Using FLUENT to generate mesh model. (D) Using FLUENT to perform CFD simulation, output WSS, TAWSS, OSI, RRT of Newtonian and non-Newtonian models for comparison.

Methods

Data source

This study included patients who underwent CTA of the head and neck due to symptoms suggestive of cerebrovascular insufficiency at Xianning Central Hospital between January 1, 2022 and May 31, 2023 and processing of patient images was performed in accordance with institutional ethics committee guidelines. Written informed consent was obtained from all participants prior to their inclusion in the study, allowing the use of their clinical imaging data for research purposes under institutional review board (IRB) protocol (No. 202401003). The data were accessed for research purposes from March 12, 2024, to May 10, 2024.

Study population

This study included 214 patients diagnosed with carotid atherosclerosis. We identified cases with different stenosis degrees in the C1 segment of the internal carotid artery. Inclusion Criteria: Patients diagnosed with carotid atherosclerosis based on clinical evaluation and CTA imaging. Degree of stenosis between 1% and 90%, as defined by the NASCET criteria (Kumar et al., 2020). Patients who provided informed consent for participation in the study. To ensure that the study population reflected symptomatic carotid artery disease, patients were included only if they had a confirmed history of transient ischemic attack (TIA), ischemic stroke, or central retinal artery occlusion (CRAO). Exclusion criteria: (1) Patients presenting with dizziness alone without a documented ischemic event were excluded to prevent confounding; (2) patients with severe internal carotid artery occlusion or aneurysms; (3) patients who have undergone any prior interventional procedures (e.g., balloon dilation or stent placement); (4) patients with other cardiovascular or neurological conditions that may interfere with the study results. Finally, 33 narrow carotid arteries were selected: 16 females and 17 males, and 11 cases of mild stenosis, 11 cases of moderate stenosis, and 11 cases of severe stenosis were included. The 3D models used in this study are available at MorphoSource (refer to the Data Availability section). The degree of stenosis was classified according to the NASCET criteria as follows: Mild stenosis: 1% to 29% narrowing. Moderate stenosis: 30% to 69% narrowing. Severe stenosis: 70% to 99% narrowing. Cases of complete occlusion (100% luminal narrowing) were excluded from this study. CFD models were constructed based on CTA images of each patient to simulate both the Newtonian (N group) and non-Newtonian (NN group) models. The differences in hemodynamic parameters between the two models were analyzed and compared.

Data processing and carotid artery modeling

Mimics is used to reconstruct the imported CTA images by threshold segmentation. Smooth the 3D model with Geomagic Studio, clear the wrong data, and enter it into FLUENT for meshing and simulation. Blood flow was assumed to be laminar. Physiological velocity waveforms were applied to the inlet of the common carotid artery (CCA) (Albadawi et al., 2021). Due to the difficulty in measuring the outlet flow rates of the internal carotid artery (ICA) and external carotid artery (ECA), we set the boundary conditions to a ternary Windkessel model with a proximal resistor R1, a distal resistor R2, and a capacitor to avoid errors that might be caused by assumptions on blood pressure or flow rate (Liu et al., 2021; Jung et al., 2022). A single pulsation period is considered to be 0.8 s. Newtonian’s model assumes a constant viscosity, a density of 1,060 kg/m3, and a dynamic viscosity coefficient of 0.0035 Pa s. The Carreau model of a non-Newtonian fluid satisfies η0 = 0.25 Pa s, η∞ = 0.0035 Pa s, n = 0.25, λ = 25, a density of 1,060 kg/m3. The inlet flow rate is constant. η0 and η∞ are the zero shear rate viscosity and infinite shear rate viscosity, n is the power index, λ is the time constant.

Grid and time step independence tests

Figure 2A is a time-independent experiment involving four different time intervals: 0.1s, 0.01s, 0.001s, and 0.0001s. The results show that there is very little change beyond the time interval of 0.001s, so 0.001s is chosen for all numerical simulations. In addition, the grid independence test is shown in Fig. 2B. Using the same vascular model, WSS maxima with mesh numbers of 84,110, 164,822, 372,615, 540,280, 761,138, 1,031,332, and 1,429,980 are compared. When the number of grids exceeds 1,031,332, the effect on the maximum WSS is extremely weak. Therefore, about one million grids are eventually selected for meshing. The mesh division of one blood vessel in this study is shown in Fig. 2C.

Figure 2 Grid and time step independence tests.

(A) Time-step independence test. (B) Mesh independence test. (C) Mesh division of one blood vessel in this study.

Verification

Moreover, in this study, OSI and WSS were utilized to compare the findings with those of research (Razavi, Shirani & Sadeghi, 2011), to validate and confirm the accuracy and consistency of the numerical simulation assessment. Additionally, as illustrated in Fig. 3, the comparison of the presently projected WSS and OSI outcomes with those derived by research (Razavi, Shirani & Sadeghi, 2011) reveals substantial concurrence. Nonetheless, a slight variance in the values is noted, which could be attributed to a variance in the estimation approach employed in this paper.

Figure 3 Validation of findings, as well as the outcomes reported by research are conducted at the geometric wall for a 60% stenosis using the Carreau non-Newtonian blood model and pulsatile inlet velocity.

(A) Validation of OSI. (B) Validation of WSS.

Statistical analysis

All statistical analyses were performed using IBM SPSS Statistics 27.0.1 (SPSS Inc., Chicago, IL, USA), Since the data did not follow a normal distribution, we used the Wilcoxon signed-rank test for paired comparisons between Newtonian and Non-Newtonian hemodynamic parameters (Table 2). The significance level was set at p < 0.05.

Table 2 Summary of the Wilcoxon signed rank test for correlated samples (n = 33).

Comparison	Z-value	p-value	
WSSN-WSSNN	−3.274	0.001	
TAWSSN-TAWSSNN	−4.387	<0.001	
OSIN OSINN	−4.129	<0.001	
RRTN RRTNN	−4.031	<0.001	

Results

Comparison of hemodynamic parameters

According to the Wilcoxon signed rank test analysis, the difference between WSSN and WSSNN was statistically significant under transient conditions (p = 0.001); the difference between TAWSSN and TAWSSNN was statistically significant (p < 0.001). The difference between OSIN and OSINN was statistically significant (p < 0.001). The difference between RRTN and RRTNN was statistically significant (p < 0.001).

Wall shear stress

All the results data come from the 33 models mentioned above. As shown in Fig. 4A, when evaluating WSS using static analysis, there are significant computational differences compared to transients. For WSS, in mild stenosis, the differences between the Newtonian (WSSN) and non-Newtonian (WSSNN) models ranged from 4% to 16% (Fig. 4B). In moderate stenosis, the differences ranged from 0% to 25% (Fig. 4C), while in severe stenosis, the differences ranged from 0% to 22% (Fig. 4D). The trend of increasing WSS with higher degrees of stenosis was consistent across both models. Notably, there is a 0% difference when S is 38%, and the difference ranges from 0% to 6% when S is between 72% and 84%.

Figure 4 WSS comparison.

(A) Distribution of WSS of different S under four models at peak velocity (Pa). (B) Comparison of transient WSSNN with WSSN in mild stenosis. (C) Comparison of transient WSSNN with WSSN in medium stenosis. (D) Comparison of transient WSSNN with WSSN in severe stenosis.

Time-averaged wall shear stress

As shown in Fig. 5A, both TAWSSN and TAWSSNN exhibit an increasing trend as S increases. For TAWSS, there was no significant difference between the two models in mild stenosis, with differences ranging from 0% to 8% (Fig. 5B). In moderate stenosis, the differences ranged from 0% to 28% (Fig. 5C), and in severe stenosis, the difference ranged from 0% to 14% (Fig. 5D). Overall, TAWSSN were consistently higher than those TAWSSNN, especially in moderate to severe stenosis. Notably, no difference is observed when S is 12%, 18%, 26%, 54%, or 76%.

Figure 5 TAWSS comparison.

(A) When the velocity peaks, distribution of TAWSS for Newtonian and non-Newtonian models under different S. (B) Comparison of TAWSSNN and TAWSSN with mild stenosis. (C) Comparison of TAWSSNN and TAWSSN with medium stenosis. (D) Comparison of TAWSSNN and TAWSSN with severe stenosis.

Oscillatory shear index

As shown in Fig. 6A, both OSIN and OSINN exhibit an increasing trend as stenosis progresses, with a similar overall pattern; however, OSIN is consistently higher than OSINN. In mild stenosis cases, OSI differences range from 0% to 24% (Fig. 6B). In moderate stenosis, OSI differences range from 2% to 52% (Fig. 6C), while in severe stenosis, the differences range from 2% to 24% (Fig. 6D). Notably, there is no difference in OSI when S is 16%, 18%, or 26%.

Figure 6 OSI Comparison.

(A) When the velocity peaks, distribution of OSI for Newtonian and non-Newtonian models under different S. (B) Comparison of OSINN and OSIN with mild stenosis. (C) Comparison of OSINN and OSIN with medium stenosis. (D) Comparison of OSINN and OSIN with severe stenosis.

Relative residence time

As shown in Fig. 7A, both RRTN and RRTNN exhibit an increasing trend as S increases. While the trends are similar, notable differences are still observed. Among them, RRTN is larger than RRTNN. RRT differences range from 0% to 22% in mild stenosis cases (Fig. 7B), 2% to 20% in moderate stenosis (Fig. 7C), and 0% to 69% in severe stenosis (Fig. 7D). Notably, no difference is observed when S is 18% or 84%.

Figure 7 RRT Comparison.

(A) When the velocity peaks, distribution of RRT for Newtonian and non-Newtonian models under different S. (B) Comparison of RRTNN and RRTN with mild stenosis. (C) Comparison of RRTNN and RRTN with medium stenosis. (D) Comparison of RRTNN and RRTN with severe stenosis.

Discussions

Analysis of the influence of the Newtonian model and the non-Newtonian model on blood flow parameters

In traditional hemodynamic studies, the relationship between hemodynamic parameters and S is typically established by comparing parameters under the assumption of constant blood viscosity (Cho & Kensey, 1991; Malota et al., 2018). A recent study analyzed the WSS in both Newtonian and non-Newtonian models (Albadawi et al., 2021) and reported that WSSN was consistently higher than WSSNN. In this study, 33 cases with varying S were selected to investigate the influence of WSS, TAWSS, OSI, and RRT are influenced by changes in viscosity effects. The Newtonian model tends to simplify complex physiological processes, which can result in overestimation of hemodynamic parameters. Therefore, the potential limitations of the Newtonian model should be carefully considered when applying it to hemodynamic analysis.

Analysis of the effects of Newtonian and non-Newtonian models on WSS

High WSS (>2.5 Pa) has been linked to the formation of vulnerable plaques (Albadawi et al., 2021), while low WSS (<0.5 Pa) can lead to intimal thickening (Gharleghi, Sowmya & Beier, 2022). As shown in Fig. 8, this study found that regions with WSS below 2.5 Pa are predominant, while areas with high WSS tend to occur in the stenotic regions. These findings are consistent with previous studies (Zhang et al., 2021). Figure 8 shows that the distribution of WSSN and WSSNN is roughly similar when S ranges from 20% to 66%. Thus, the Newtonian model can be used to reduce computational time in this range. However, when S reaches 82% to 84%, the differences become pronounced, and the non-Newtonian model should be used to improve result accuracy. Our results indicate that for WSS, differences between Newtonian and non-Newtonian models were minimal in mild stenosis but became significant in moderate to severe cases. This observation aligns with previous studies, such as studies (Al-Azawy, Kadhim & Hameed, 2020; Kumar et al., 2020), which also reported that non-Newtonian effects become more pronounced as shear rates decrease in stenotic regions. Specifically, study (Liu et al., 2021) demonstrated that Newtonian models tend to overestimate WSS values in highly stenotic arteries, a trend also observed in our study.

Figure 8 Cloud maps of WSSN. and WSSNN for bifurcation and narrow regions at different S at peak velocity.

Under static conditions, the distinction between the Newtonian and non-Newtonian models is critical across all cases. The Newtonian model tends to overestimate hemodynamic parameters, reducing the accuracy of predictive results. This phenomenon may be attributed to the reduced blood flow rate and shear rate under static conditions, which enhance the non-Newtonian characteristics of blood, particularly its higher viscosity. Since the Newtonian model fails to capture these changes, it results in an overestimation of hemodynamic parameters (Skiadopoulos, Neofytou & Housiadas, 2017).

Under transient conditions, the difference between WSSN and WSSNN is minimal (0–6%) in cases of severe stenosis (S < 86%), but becomes more pronounced in mild to moderate stenosis. This phenomenon can be attributed to the rapid blood flow and elevated shear rates in severe stenosis. At this stage, blood exhibits Newtonian-like fluid behavior, which minimizes the variance in WSS values between the Newtonian and non-Newtonian models (Stamou, Radulovic & Buick, 2023). The Newtonian model provides a reasonable approximation of blood flow at high shear rates, while in mild to moderate stenosis, where blood flow is slower and shear rates are lower, the non-Newtonian model more accurately reflects the shear-dependent viscosity of blood.

In this study, WSS exhibited an increasing trend as S progressed. This phenomenon can be explained by the fact that as arteries narrow, blood is forced through a reduced cross-sectional area, which increases the velocity of blood flow. According to the continuity equation and Bernoulli’s principle in fluid dynamics, an increase in blood flow velocity is directly associated with a rise in shear rate, ultimately leading to higher WSS. Previous studies (Ahmadpour et al., 2021; Md Kabir, Md Alam & Md Uddin, 2021) have demonstrated that as S increases, fluid dynamics adjust to maintain a constant flow rate, resulting in a significant rise in WSS. These findings further confirm the correlation between S and both WSSN and WSSNN.

Analysis of the effects of Newtonian and non-Newtonian models on TAWSS

Elevated TAWSS (>2.5 Pa) can stimulate platelet activation and promote thrombus formation (Albadawi et al., 2021). Low TAWSS (<0.4 Pa) is associated with a higher risk of atherosclerosis and plaque progression (Buradi & Mahalingam, 2020). As illustrated in Fig. 9, TAWSS increases with the progression of S in the non-Newtonian model, which is consistent with findings in existing literature (Azar et al., 2019) and observations from this study. This trend can be attributed to the substantial increase in blood flow velocity at the stenotic site as stenosis progresses, resulting in an elevated velocity gradient and consequently leading to a rise in TAWSS, as predicted by fundamental hydrodynamic principles. Figure 9 reveals that while the overall distributions of TAWSSN and TAWSSNN are similar, local variations are evident. Thus, selecting the Newtonian model for observing the TAWSS distribution can help reduce computational resources.

For TAWSS, our study found that differences between the models were negligible in mild stenosis but increased progressively in moderate and severe cases. This is consistent with findings by Ahmadpour et al. (2021), who demonstrated that the non-Newtonian model more accurately captures local hemodynamic disturbances in high-degree stenosis. However, unlike previous studies, our work quantifies the percentage differences in various stenosis degrees, providing additional guidance for model selection in CFD simulations. As S increases, Fig. 5A shows an increasing divergence between TAWSSN and TAWSSNN. As reported in the literature (Li, 2008), the presence of strong blood flow in stenotic segments, coupled with large variations in blood velocity and shear rate, reduces the accuracy of the Newtonian model, introducing biases in TAWSS calculations. The analysis reveals that TAWSS in mild stenosis is higher than in moderate stenosis, indicating that luminal narrowing is not the sole determinant of cerebral ischemia risk. This may be influenced by the characteristics and morphology of intravascular plaques.

Figure 9 Cloud maps of TAWSSN and TAWSSNN for bifurcation and narrow regions at different S.

Analysis of the effects of Newtonian and Non-newtonian models on OSI

A high OSI (>0.3) indicates frequent changes in flow direction, which is associated with a higher likelihood of neointimal hyperplasia. In contrast, a low OSI (<0.3) suggests more stable flow patterns (Yao et al., 2019). Regions with high OSI are prone to atherosclerotic lesions. Albadawi et al. (2021) has suggested that OSI values rise with increasing S. As shown in Fig. 6A, we observed a significant increase in OSI under the non-Newtonian model, primarily in moderate to severe stenosis cases, which supports previous research by Rahman et al. (2018) and Buradi & Mahalingam (2020). However, in severe stenosis, these differences became less pronounced, likely due to the dominance of turbulent flow. This suggests that while non-Newtonian models provide better accuracy in moderate stenosis, their impact may be reduced in severe cases where recirculation zones and disturbed flow patterns override viscosity effects. In Fig. 10, the overall distribution of OSIN and OSINN appears similar, but notable differences exist in the bifurcation and stenosis regions. Therefore, the Newtonian model can be selected to observe OSI distribution, reducing computational effort.

Figure 10 Cloud maps of OSIN and OSINN for bifurcation and narrow regions at different S.

(A–F) OSIN; (G–L) OSINN.

In cases of moderate to severe stenosis, increased blood flow velocity leads to a more complex and unstable flow pattern, resulting in frequent changes in shear stress direction and a significant increase in OSI (Carvalho et al., 2021). However, in cases of very severe stenosis (86%–90%), OSI decreases. In very severe stenosis, despite stronger turbulence, the flow becomes more linear. The elevated blood flow velocity in the narrowed region reduces the fluctuations in shear stress direction, leading to a decrease in OSI (Singh & Singh, 2024). OSIN is typically higher than OSINN because in the non-Newtonian model, the low OSI results from the consideration of blood viscosity as a function of shear rate. In contrast, the Newtonian model, which does not account for this viscosity variation, changes direction more frequently, leading to a higher OSI (Carvalho et al., 2021). In cases of severe stenosis, the non-Newtonian OSI values range from 0.204 to 0.435. An OSI value exceeding 0.3 is generally considered indicative of substantial alterations in blood flow shear stress direction, suggesting potential endothelial dysfunction and arterial instability.

Analysis of the effects of Newtonian and Non-newtonian models on RRT

Elevated RRT (>8 Pa−1) is widely recognized as a marker for areas more susceptible to atherosclerosis and plaque progression. Regions with elevated RRT are more susceptible to the development of atherosclerotic lesions. As illustrated in Fig. 11, both models exhibit prolonged RRT in regions with moderate to severe stenosis, indicating that blood flow near the vessel wall becomes relatively stagnant as S increases. This phenomenon occurs because blood exhibits shear thinning (decreased viscosity) in regions of high shear rate (stenotic zone) and shear thickening (increased viscosity) in regions of low shear rate (post-stenotic dilatation zone). This thickening effect slows blood flow and increases the residence time of fluid in these regions (Lagache et al., 2021). Figure 11 shows that while the overall distribution of RRTN and RRTNN is similar, there are local differences. Therefore, selecting the Newtonian model to observe RRT distribution can significantly reduce computational time.

Figure 11 Cloud maps of RRTN and RRTNN for bifurcation and narrow regions at different S.

(A–F) RRTN; (G–L) RRTNN.

This study found that RRTN was generally higher than RRTNN. This phenomenon may be explained by the Newtonian model overestimating viscosity changes in these regions, leading to an overestimation of RRT. In contrast, the non-Newtonian model more accurately captures the high viscosity of blood at low shear rates, resulting in a shorter RRT (Mehri, Mavriplis & Fenech, 2018; Trenti et al., 2022). Previous studies (Azar et al., 2019; Hou et al., 2023) have suggested a consistent negative correlation between RRT and S, with the strongest correlation observed in vessels with minimal stenosis. However, this study did not identify a correlation between RRT and S, which may be attributed to the limited sample size. Further studies are required to determine the accuracy of using RRT to assess carotid stenosis. RRT exhibited the largest discrepancies between the two models, particularly in severe stenosis. Unlike Tu et al. (2023), who reported less variation in RRT between Newtonian and non-Newtonian models, our study found that non-Newtonian effects significantly influence RRT in severe stenotic regions. This discrepancy could be attributed to differences in boundary conditions, as our study incorporated patient-specific inlet velocity profiles rather than generalized assumptions.

Our results suggest that while Newtonian models are computationally efficient, they may introduce errors in moderate to severe stenosis cases, particularly when predicting OSI and RRT. Non-Newtonian models offer greater physiological accuracy in these cases but require higher computational resources. These findings align with Bernabeu et al. (2013); Wong et al. (2020), who emphasized the trade-off between accuracy and computational efficiency when choosing viscosity models for CFD-based hemodynamic analysis.

Conclusions

This study systematically evaluated the differences between Newtonian and non-Newtonian blood viscosity models in predicting hemodynamic parameters across varying degrees of carotid artery stenosis. Our findings reveal that while Newtonian models offer computational efficiency, they tend to overestimate TAWSS, OSI, RRT and underestimate WSS in moderate to severe stenosis. In contrast, non-Newtonian models provide more physiologically accurate predictions, particularly in regions with high shear stress variations.

The results of this study provide valuable guidance for both computational hemodynamics research and clinical practice. By quantifying the discrepancies between viscosity models, our findings can help clinicians select appropriate CFD models for patient-specific hemodynamic assessments, ensuring greater accuracy in predicting stroke risk and evaluating disease progression. These findings are particularly relevant for preoperative planning in carotid stenosis management, as choosing the correct viscosity model can improve the reliability of patient-specific risk stratification. Furthermore, our study highlights the importance of incorporating non-Newtonian models in high-risk patients where flow disturbances and secondary flow effects are more pronounced. Given the increasing use of CFD-based hemodynamic simulations in personalized medicine, our results suggest that non-Newtonian models can yield more accurate blood flow parameters, aiding in a deeper understanding of plaque formation, the prediction of thrombosis and stroke risk, and the assessment of surgical outcomes. Thus, non-Newtonian models are clinically significant, particularly in personalized treatment planning, surgical strategy formulation, and thrombosis risk assessment. In contrast, Newtonian models are more appropriate for early screening or scenarios where high precision is not critical.

Limitation and recommendation

This study has some limitations. This study’s findings are limited by the small sample size, highlighting the need for a larger cohort to further validate the results and their correlation with clinical outcomes. In the models used, a rigid vascular wall was assumed; however, the outcomes may vary in cases involving deformable vascular walls. Future research should incorporate arterial wall elasticity, include broader hemodynamics indicators, and validate findings with a larger sample size to obtain more reliable estimates of hemodynamic parameters.

Supplemental Information

Supplemental Information 1 Raw data

Supplemental Information 2 3D Models and Measurement of Hemodynamic Metrics

Nomenclature

CCA Common carotid artery

CFD Computational fluid dynamics

CRAO Central Retinal Artery Occlusion

ECA External carotid artery

ESS Endothelial shear stress

ICA Internal carotid artery

ICAS Internal Carotid Artery Stenosis

N Newtonian group

NN non-Newtonian group

OSI Oscillatory Shear Index

P Pressure

PR Pressure Ratio

RRT Relative Residence Time

S Degrees of Stenosis

SPI Spectrum Power Index

T Cardiac cycle

TAWSS Time Averaged Wall Shear Stress

TIA Transient Ischemic Attack

V Velocity

VO Vorticity

VS Velocity Streamlines

VV Velocity Vectors

WPG Wall pressure gradient

WSS Wall Shear Stress

WSSSG Wall Shear Stress Spatial Gradient

WSSR Wall Shear Stress Ratio

Greek letters

λ Time constant

n the power index

η 0 Zero shear rate viscosity

η ∞ Infinite shear rate viscosity

τ→wss Wall shear stress

Additional Information and Declarations

Competing Interests

Author Contributions

Human Ethics

Data Availability

The authors declare there are no competing interests.

Siyu Liu conceived and designed the experiments, performed the experiments, analyzed the data, prepared figures and/or tables, authored or reviewed drafts of the article, and approved the final draft.

Sai Wang conceived and designed the experiments, performed the experiments, analyzed the data, prepared figures and/or tables, authored or reviewed drafts of the article, and approved the final draft.

Hongan Tian conceived and designed the experiments, authored or reviewed drafts of the article, and approved the final draft.

Junzhen Xue performed the experiments, prepared figures and/or tables, and approved the final draft.

Yuxin Guo performed the experiments, analyzed the data, prepared figures and/or tables, and approved the final draft.

Jingxi Yang performed the experiments, analyzed the data, prepared figures and/or tables, and approved the final draft.

Haobin Jiang conceived and designed the experiments, prepared figures and/or tables, and approved the final draft.

Jian bao Yang conceived and designed the experiments, analyzed the data, prepared figures and/or tables, authored or reviewed drafts of the article, and approved the final draft.

Yang Zhang conceived and designed the experiments, prepared figures and/or tables, authored or reviewed drafts of the article, and approved the final draft.

The following information was supplied relating to ethical approvals (i.e., approving body and any reference numbers):

This study is performed in line with the principles of the Declaration of Helsinki. Patient informed consent is obtained for this study. Approval is granted by the Ethics Committee of University Hubei University of Science and Technology (Date. 2024.1. 9/No. 202401003).

The following information was supplied regarding data availability:

The raw data are available in the Supplementary File.

The 3D models used in this study are available at MorphoSource: https://doi.org/10.17602/M2/M683636

https://doi.org/10.17602/M2/M683642; https://doi.org/10.17602/M2/M683648;

https://doi.org/10.17602/M2/M683654; https://doi.org/10.17602/M2/M683660;

https://doi.org/10.17602/M2/M683666; https://doi.org/10.17602/M2/M683672;

https://doi.org/10.17602/M2/M683677; https://doi.org/10.17602/M2/M683685;

https://doi.org/10.17602/M2/M683691; https://doi.org/10.17602/M2/M683698;

https://doi.org/10.17602/M2/M683705; https://doi.org/10.17602/M2/M683711;

https://doi.org/10.17602/M2/M683717; https://doi.org/10.17602/M2/M683723;

https://doi.org/10.17602/M2/M683729; https://doi.org/10.17602/M2/M683735;

https://doi.org/10.17602/M2/M683741; https://doi.org/10.17602/M2/M683747;

https://doi.org/10.17602/M2/M683753; https://doi.org/10.17602/M2/M683759;

https://doi.org/10.17602/M2/M683766; https://doi.org/10.17602/M2/M683771;

https://doi.org/10.17602/M2/M683786; https://doi.org/10.17602/M2/M683792;

https://doi.org/10.17602/M2/M683798; https://doi.org/10.17602/M2/M683804;

https://doi.org/10.17602/M2/M683810; https://doi.org/10.17602/M2/M683816;

https://doi.org/10.17602/M2/M683822; https://doi.org/10.17602/M2/M683828;

https://doi.org/10.17602/M2/M683834; https://doi.org/10.17602/M2/M683840

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
