# Peer review of "Comparison of blood viscosity models in different degrees of carotid artery stenosis"

_PeerJ, doi:10.7717/peerj.19336_

## Round 0.1 · original submission · Major Revisions

Please address concerns pf both reviewer and revise manuscript accordingly.

·

Basic reporting

First of all, the English language should be checked throughout the entire manuscript. There are sentences that lose their meaning as they are read or that do not make sense. Additionally, the content of the manuscript should be easy to read and follow, with simpler sentences rather than overly elaborate phrases.
The references cited in the manuscript text are consistent with the subject of the article.
Unfortunately, some of the figures presented in the manuscript (Figure 3 – Figure 7) do not have satisfactory resolution, which is why this aspect should be improved.
The introduction is overly elaborate, causing the core message to be lost.

Experimental design

The manuscript should clearly include the inclusion and exclusion criteria of the presented study. Additionally, the percentage definitions of mild, moderate, and severe stenosis should be established. The entire Materials and Methods section is difficult to follow due to excessive details (such as formulas and Morphosource), which should be included in the supplementary materials to avoid complicating the readability of the article. Citations should be avoided in this section.
Authors should define clearly the research question and they should state how the study contributes to the actual research in the field.

Validity of the findings

The Results section should not include citations or explanations, as these belong in the Discussion section. The Results should fully and comprehensively describe the findings of the present study, including only information related to the patient cohort. Definitions used in the Results section should be moved to the Introduction. The Discussion is well-conducted and relevant to the manuscript; however, the presented study should also be included, making comparisons with the cited references.
The conclusions need to be reconsidered. They should not restate the information already presented in the article but should emphasize the significance of the results for current research and how these findings can contribute to improving medical practice. Specifically, how do these results help patients and clinicians, and what added value do they bring to the existing knowledge about carotid stenosis?

Additional comments

No additional comments.

Reviewer 2 ·

Basic reporting

1. The introduction is good but lengthy. Please consider narrowing it down to two or three paragraphs with only the most relevant information.
Minor:
2. Please fix typographical errors.
3. I would suggest representing stenosis in percentages instead of proportions.

Experimental design

1. If a CTA of the head and neck was performed due to dizziness, it is unlikely to indicate symptomatic carotid artery disease. Please mention whether these patients had TIA or ischemic stroke or central retnal artery occlusion.
2. Please specify what constitutes "severe high-grade stenosis" as mentioned in the methods section. Is it greater than 70%? Authors used the NASCET criteria to determine the degree of stenosis, however, did not specify the definition of severe stenosis. Additionally, stenosis is different than occlusion
3. Please explain why statistical testing was not performed.

Validity of the findings

Please perform statistical testing to compare Newtonian and non Newtonian hemodynamic parameters.

---

## Round 0.2 · accepted · Accept

All issues pointed out by the reviewers were adequately addressed and the revised version is acceptable now.

·

Basic reporting

The authors made significant changes to the article that improve its accuracy and provide a better understanding of what they aimed to achieve. The references are consistent with the article and the information published in it. The raw data provides additional information for understanding the conducted study. The results are relevant to the medical literature.

Experimental design

The article presents original primary research that aligns well with the Aims and Scope of the journal, making a valuable contribution to the field. The research question is clearly defined, relevant, and meaningful, addressing an identified knowledge gap in the current literature. The authors have effectively outlined how their work fills this gap, which adds depth to the research's significance.

The investigation is conducted with a high level of rigor, adhering to both technical and ethical standards, ensuring the reliability and validity of the findings. The methods are described in sufficient detail, allowing for replication of the study. This transparency enhances the robustness of the research and supports its reproducibility, which is crucial for advancing scientific knowledge. Overall, the study meets the journal’s criteria for high-quality research and provides meaningful insights to the field.

Validity of the findings

The impact and novelty of the study are well-positioned to make a meaningful contribution to the field. The authors have effectively addressed the research question, and highlighting the rationale and benefits to the literature would further emphasize the study's significance. Encouraging replication, with a clear rationale for its importance, would strengthen the overall impact of the research.

All underlying data have been provided, and they are robust, statistically sound, and carefully controlled, which enhances the reliability of the findings. The authors' transparency in presenting these data allows for further validation and supports the study's scientific rigor.

The conclusions are well-articulated, closely linked to the original research question, and appropriately supported by the study's results. This ensures the clarity and relevance of the conclusions, highlighting the contribution of the research to the field in a focused and impactful way.

Additional comments

Nothing

Reviewer 2 ·

Basic reporting

The authors did a great job in addressing my concerns.

Experimental design

No concerns

Validity of the findings

The authors incorporated my suggestions.